# Tortuosity of the Internal Carotid Artery and Its Clinical Significance in the Development of Aneurysms

**DOI:** 10.3390/jcm8020237

**Published:** 2019-02-12

**Authors:** Kornelia M. Kliś, Roger M. Krzyżewski, Borys M. Kwinta, Krzysztof Stachura, Jerzy Gąsowski

**Affiliations:** 1Faculty of Medicine, Jagiellonian University Medical College, 31-008 Kraków, Poland; korneliakli@gmail.com; 2Faculty of Computer Science, Electronics and Telecommunications, AGH University of Science and Technology, 30-059 Kraków, Poland; 3TENSOR—Team of NeuroSurgery-Oriented Reaserch, Jagiellonian University Medical College, 31-008 Kraków, Poland; bmkwinta@gmail.com (B.M.K.); jerzy.gasowski@gmail.com (J.G.); 4Department of Neurosurgery and Neurotraumatology, Jagiellonian University Medical College, 31-503 Kraków, Poland; kkstach@poczta.onet.pl; 5Department of Internal Medicine and Gerontology, Jagiellonian University Medical College, 31-531 Kraków, Poland

**Keywords:** intracranial aneurysm, tortuosity, internal carotid artery

## Abstract

Tortuosity of blood vessels is a common angiographic finding that may indicate systemic disease and can be correlated with vascular pathologies. In this work, we determined whether patients with and without internal carotid artery (ICA) aneurysm presented with differences in its tortuosity descriptors. We retrospectively analysed data of 298 patients hospitalized between January 2014 and June 2018. For each patient’s imaging data, we extracted a curve representing the ICA course and measured its Relative Length (RL), Sum of Angle Metrics (SOAM), Product of Angle Distance (PAD), Triangular Index (TI), and Inflection Count Metrics (ICM). We found that patients with an ICA aneurysm had significantly lower RL (0.46 ± 0.19 vs. 0.51 ± 0.17; *p* = 0.023) and significantly higher SOAM (0.39 ± 0.21 vs. 0.32 ± 0.21 *p* = 0.003), PAD (0.38 ± 0.19 vs. 0.32 ± 0.21; *p* = 0.011), TI (0.30 ± 0.11 vs. 0.27 ± 0.14; *p* = 0.034), and ICM (0.30 ± 0.16 vs. 0.22 ± 0.12; *p* < 0.001). We found that that patients who presented with a subarachnoid hemorrhage had significantly higher PAD (0.46 ± 0.22 vs. 0.35 ± 0.20; *p* = 0.024). In conclusion, higher tortuosity of ICA is associated with ICA aneurysm presence.

## 1. Introduction

Tortuosity of blood vessels is a common angiographic finding that might indicate systemic diseases, such as hypertension or diabetes mellitus [1,2], and can be correlated with vascular pathologies [3,4]. It can also increase with age [5]. Tortuosity is most commonly analysed in retinal and coronary vessels; however, it can be found in a vast majority of blood vessels [6]. In terms of brain vasculature, tortuosity is associated with Moyamoya disease [7] and presence of atherosclerosis [8]. Sprangler et al. found a correlation between hypertension and white matter arterioles [9].

A few mechanisms may be linked to an increase in tortuosity, the first of which are mechanical factors of blood flow, such as elevated blood pressure [2] or reduced axial tension [6]. Another factor that might promote tortuosity is the weakening of arterial walls, resulting either from elastin degradation or abnormal deposits within vessel walls [10]. Tortuosity could also result from an increase in blood flow [2].

As tortuosity promotes hemodynamic changes in blood flow, it can lead to the development of aneurysms. Such association was found in terms of the aorta [11] and splenic artery [12], as well as in brain arteries, such as the internal carotid artery (ICA) [13], basilar artery (BA) [14], middle cerebral artery (MCA) [15], and vertebral artery [16]. Our previous study suggested that tortuosity of the anterior cerebral artery might play a role in anterior communicating artery aneurysm rupture [17]. However, some of the authors who analysed tortuosity used subjective methods based on visual appearance [13]. Therefore, we decided to perform a computer-aided analysis of ICA and objectively determine whether there is a difference in mathematical tortuosity factors between groups of patients with and without ICA aneurysm.

## 2. Experimental Section

We retrospectively analysed the data of 298 patients hospitalized between January 2014 and June 2018 who underwent digital subtraction angiography (DSA) due to suspicion of an intracranial aneurysm. Our study group included 149 patients with ICA aneurysm and 149 patients in the control group without ICA aneurysm, matched for age (±3 years) and risk factors (hypertension, diabetes mellitus, and smoking). Aneurysm presence was confirmed by DSA. Patients with multiple aneurysms, ICA aneurysms located in extracranial segment, suspicion of intracranial vasospasm, connective tissue disorders, or patients who did not provide informed consent to participate in the study were excluded. For each patient, we obtained their medical history from their medical records, including previous and current diseases and medications, as well as aneurysm data such as its size and exact location. Mirror aneurysms were defined as both-sided aneurysms on the same segment of ICA. We obtained patients’ imaging data prior to surgical or endovascular treatment. The study protocol was approved by a local bioethical committee and all patients provided informed consent. Database and source code of the used software are available to readers upon request. The primary endpoint for our study was to determine an association between tortuosity of intracranial segments of ICA and presence of ICA aneurysm. Secondary endpoints included determining a possible association between ICA tortuosity and common risk factors for aneurysm development, as well as between ICA tortuosity and ICA measurements.

Methods of artery tracking and details about the software used for this study were described in our previous work [15]. For each patient’s DSA, we extracted a curve representing the ICA intracranial course (C2–C7 segments) and measured its Relative Length (RL), Sum of Angle Metrics (SOAM), Product of Angle Distance (PAD), Triangular Index (TI), and Inflection Count Metrics (ICM). The formulas for each descriptor calculation are presented in Figure 1. We measured diameters of ICA segments C6 and C7 as well as the mean diameter of the entire ICA obtained from its three measurements—2 mm from each end and in the middle.

The database management and statistical analysis were performed with RStudio version 8.5 for Windows (RStudio, Inc., Boston, MA, USA). We used the Shapiro-Wilk test to assess normality. For comparisons of continuous variables, we used the *t*-test for normally distributed variables, and the Mann-Whitney *U* test for non-normally distributed variables. We used the *χ*^2^ test for dichotomous variables. To assess correlation between continuous variables, we used Pearson’s or Spearman’s correlation tests for normally and non-normally distributed variables, respectively. We express continuous variables as a mean (standard deviation). To find factors independently associated with the presence of an ICA aneurysm, we employed logistic regression analysis with and without adjustment for possible confounders. All significance tests are two-tailed and a *p*-value < 0.05 was considered significant.

## 3. Results

### 3.1. Study Group Characteristics

Our study group included 298 patients and 227 (76.17%) were women. The mean age of the study group was 57.48 ± 12.66 years. Among the patients in the study group, the mean diameter of the C6 segment was 3.88 ± 0.86 mm, the mean diameter of the C7 segment was 2.92 ± 0.72 mm, the mean diameter of MCA was 1.93 ± 0.57 mm, and the mean ACA diameter was 1.75 ± 0.48 mm. In terms of tortuosity, average RL was 0.48 ± 0.19, average SOAM was 0.36 ± 0.21, average PAD was 0.35 ± 0.20, average TI was 0.29 ± 0.13, and average ICM was 0.26 ± 0.14. Among patients with an ICA aneurysm, the most common location of the aneurysm was the C7 segment (49.62%), then the C6 segment (39.10%), C4 segment (9.02%), and C5 segment (2.26%). Eighty-four (56.38%) aneurysms were located on the right side, 78 (52.35%) aneurysms were located on left side, and 21 aneurysms (14.09%) were mirror aneurysms. The mean size of the aneurysm dome was 7.19 ± 4.89 mm and the mean size of the aneurysm neck was 2.98 ± 1.18 mm.

### 3.2. Risk Factors for Aneurysm Presence

Our study showed that ICA aneurysms were more common in women (83.89% vs. 68.46%; *p* = 0.002). Subarachnoid hemorrhage was also more common in women than men (9.40% vs. 2.68%; *p* = 0.015). Women were also significantly less likely to have a history of ischemic stroke (4.70% vs. 14.09%; *p* = 0.005). Additionally, for patients with ICA aneurysm more commonly consumed acetylsalicylic acid (26.17% vs. 12.08%; *p* = 0.002), AT_2_-blockers (2.68% vs. 0%; *p* = 0.044), and statins (8.72% vs. 2.68%; *p* = 0.025). We found that the C6 segment diameter was significantly smaller in these patients (3.77 ± 0.88 mm vs. 3.99 ± 0.82 mm; *p* = 0.023). In terms of tortuosity descriptors, patients with an ICA aneurysm presented with a significantly lower RL (0.46 ± 0.19 vs. 0.51 ± 0.17; *p* = 0.023), and significantly higher SOAM (0.39 ± 0.21 vs. 0.32 ± 0.21 *p* = 0.003), PAD (0.38 ± 0.19 vs. 0.32 ± 0.21; *p* = 0.011), TI (0.30 ± 0.11 vs. 0.27 ± 0.14; *p* = 0.034), and ICM (0.30 ± 0.16 vs. 0.22 ± 0.12; *p* < 0.001) (Table 1, Figure 2).

### 3.3. Association of Risk Factors with Tortuosity

We found that female patients had a significantly higher SOAM (0.37 ± 0.21 vs. 0.48 ± 0.18; *p* = 0.028), PAD (0.37 ± 0.20 vs. 0.30 ± 0.20; *p* = 0.016), TI (0.30 ± 0.13 vs. 0.25 ± 0.10; *p* = 0.006), and ICM (0.27 ± 0.15 vs. 0.23 ± 0.13; *p* = 0.046). Our study showed that patients with a history of subarachnoid hemorrhage had significantly higher PAD (0.46 ± 0.22 vs. 0.35 ± 0.20; *p* = 0.024) (Table 2). However, there were no significant differences in terms of tortuosity between patients with and without risk factors such as hypertension, diabetes mellitus, smoking, history of myocardial infarction, or history of ischemic stroke.

### 3.4. Additional Findings

A significant negative correlation between RL and mean ICA diameter (*r* = −0.187; *p* = 0.001) was found. A significant positive correlation of TI (*r* = 0.534; *p* = 0.049) and ICM (*r* = 0.773; *p* = 0.001) with time after subarachnoid hemorrhage (SAH) was found among patients who had a history of SAH. However, there was no significant correlation between tortuosity descriptors and age or other artery measurements. There was also no significant difference in terms of tortuosity between patients with different aneurysm locations. Our study showed that patients with mirror aneurysms had a significantly lower TI (0.31 ± 0.11 vs. 0.25 ± 0.09; *p* = 0.026) than other patients with ICA aneurysms (Table 3).

## 4. Discussion

Our study showed that patients with ICA aneurysms have significantly higher ICA tortuosity. Similar results were obtained by Labeyrie et al. [13]; however, they determined tortuosity based on visual appearance and not measurements. Therefore, tortuosity might have been assessed subjectively. Our study objectively proves this association in terms of all tortuosity descriptors and, therefore, in all types of tortuosity. In our previous study [15], a similar correlation was shown in the middle cerebral artery for RL, PAD, TI, and ICM. However, we achieved contradictory results for SOAM. Association between higher tortuosity and intracranial aneurysm was shown in terms of BA [14] and VA [16], as well as in terms of the splenic artery [12] and aorta [11]. One of the explanations for such correlation could be changes in hemodynamics caused by increase in tortuosity. We showed that more tortuous coronary vessels are characterized by lower wall shear stress (WSS) and prolonged relative residence time (RRT) [18]. Association of lower WSS with aneurysm development was shown in other studies [19]. Both lower WSS and prolonged RRT can promote an inflammatory response in the arterial wall and therefore lead to atherosclerotic changes [18,20]. This mechanism is confirmed by a study by Kim et al. [8], which showed that an increase in tortuosity of MCA is related to cerebral atherosclerosis. Weakening of the arterial wall caused by atherosclerotic plaques could result in aneurysm development [21]. Additionally, lower WSS promotes matrix metalloproteinases activation [22], which also plays a role in the formation of aneurysms [23]. Another explanation for the correlation shown in our study could be the fact that tortuosity can be caused by elevated blood pressure and blood flow [2], weakening of the arterial wall due to elastin degradation [10], and reduced axial tension [6]. All these factors could lead to aneurysm development.

Our study showed a significant association between the female sex and tortuosity. Similar findings were reported in terms of women older than 60 years old [24]. Higher vessel tortuosity in female patients was also found in our previous study concerning the middle cerebral artery [15], and in a study performed by Chiha et al. regarding coronary arteries [25]. Our findings might be explained by the fact that the anatomy of the circle of Willis differs between men and women. Horokoshi et al. found that type P of the circle of Willis is more common among women [26]. In that type of anatomy, the P1 segment of the posterior cerebral artery is missing, which might result in increased blood flow in the ICA. Lindekleiv et al. showed that women had higher blood flow velocities and wall shear stress (WSS) at the ICA bifurcation [27]. All these findings might explain the increase in tortuosity in the ICA. Higher tortuosity of ICA among female patients also explains the higher prevalence of ICA aneurysms among these patients, which was presented both in this and other studies [28,29].

Another interesting finding of our study is the higher ICA tortuosity in patients who had SAH. Tortuosity is correlated with the time after SAH onset. Such findings suggest that SAH promotes arterial remodeling, which might lead to an increase in tortuosity. SAH is related to an increase in matrix metalloproteinase 9 levels [30], which might promote extracellular matrix remodeling [31]. Degradation of extracellular matrix may be one of the causes of tortuosity increase [32]. SAH also causes an increase in proangiogenic markers, such as vascular endothelial growth factor (VEGF) [33] and tumor necrosis factor alpha (TNFα) [34], which can lead to an increase in the tortuosity of cerebral vessels [35]. As all these changes were detected a short time after SAH onset, our findings suggest that they promote long-term remodeling of cerebral vessels.

Since our study has a retrospective nature, it remains unclear whether aneurysm presence could promote an increase in its feeding artery tortuosity. In this case, the correlation between the presence on an aneurysm and tortuosity could be interpreted inversely. Lee et al. showed that aneurysmal arteries are characterized by decreased critical buckling pressure [36], which can lead to an increase in tortuosity [6]. However, another result of our study—the significantly lower ICA tortuosity among patients with mirror aneurysm—might not support that finding. Mirror aneurysms are characterized by a different etiology. They might result from congenital factors such as defects in cell migration during angiogenesis, which causes alterations of blood flow [37]. Such aneurysms exhibit a genetic predisposition and tend to rupture earlier in life [37]. Therefore, as mirror aneurysms are unlikely to be caused by an increase in tortuosity, our findings suggest that aneurysm presence might not meaningfully influence the tortuosity of its feeding artery. However, due to a small number of patients with mirror aneurysms who participated in our study, such conclusions demand further investigation.

A significant negative correlation between RL and ICA diameter was also observed in this study. A similar correlation was found in terms of coronary arteries [38]. The fact that a larger artery diameter could be caused by its wall weakening and increased blood flow [10] explains such correlation.

## 5. Conclusions

Higher tortuosity of ICA is associated with aneurysm presence in any of its intracranial segments. Tortuosity is increased in female patients and patients with a history of SAH. ICA tortuosity is significantly lower among patients with mirror aneurysms, positively correlated with time after SAH, and negatively correlated with ICA diameter.

Our study was mainly limited by the study group size and the fact that our control group consisted of patients with aneurysms located in other arteries than ICA. The inability to determine influence of aneurysm presence on its feeding artery tortuosity can be viewed as another limitation. This was addressed in our findings.

## Figures and Tables

**Figure 1 jcm-08-00237-f001:**
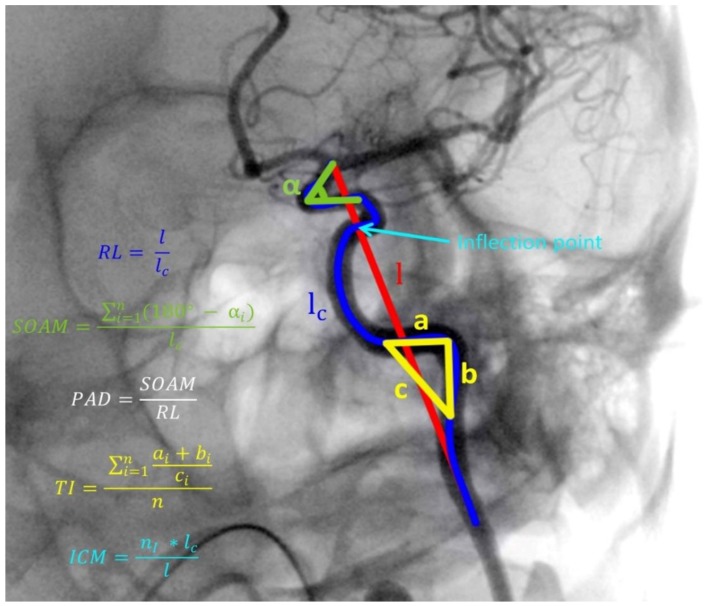
Illustration of internal carotid artery (ICA) tracking and tortuosity descriptors calculation. RL, Relative Length; SOAM, Sum of Angle Metrics; PAD, Product of Angle Distance; TI, Triangular Index; ICM, Inflection Count Metrics; *n*, number of angles on ICA course; an d*n_i_*, number of inflection points on ICA course. Marked angle, triangle, and inflection point are exemplary and are applicable to all parts of the artery. l, length of straight line between starting and ending point of analyzed artery segment; l_c_, absolute length of analyzed artery segment; a, b and c, sides of triangle constructed on angle of analyzed artery segment; α, angle of analyzed artery segment.

**Figure 2 jcm-08-00237-f002:**
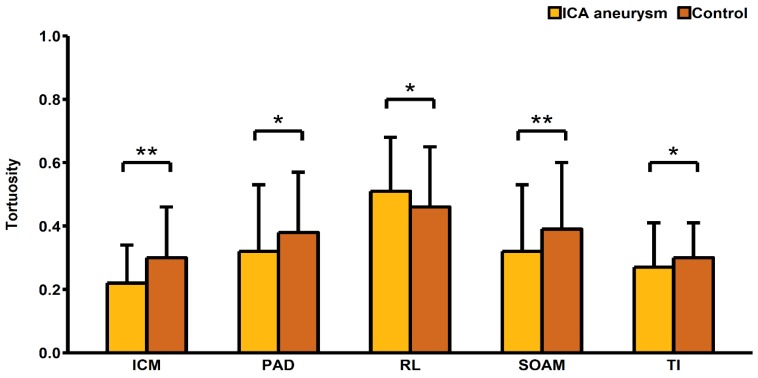
Comparison of tortuosity descriptors between patients with and without internal carotid artery aneurysm. ICA, internal carotid artery; RL, Relative Length; SOAM, Sum of Angle Metrics; PAD, Product of Angle Distance; TI, Triangular Index; ICM, Inflection Count Metrics; * *p*-value < 0.05; ** *p*-value < 0.01.

**Table 1 jcm-08-00237-t001:** Comparison of risk factors and tortuosity descriptors between patients with and without internal carotid artery aneurysm.

Variable	ICA Aneurysm (*n* = 149)	No ICA Aneurysm (*n* = 149)	*p*-Value
Female sex (%)	83.89 (125)	68.46 (102)	0.002
Age (years) ± SD	57.49 ± 12	57.48 ± 13.32	0.993
**Risk Factors**			
Diabetes mellitus (%)	9.40 (14)	12.08 (18)	0.454
Smoking (%)	12.75 (19)	12.75 (19)	0.999
Hypertension (%)	47.65 (71)	47.65 (71)	0.999
Alcoholism (%)	0 (0)	4.70 (7)	0.007
Ischemic heart disease (%)	1.34 (2)	3.36 (5)	0.251
History of heart attack (%)	0.67 (1)	2.68 (4)	0.176
History of ischemic stroke (%)	4.70 (7)	14.09 (21)	0.005
History of subarachnoid hemorrhage (%)	9.40 (14)	2.68 (4)	0.015
Atrial fibrillation (%)	1.34 (2)	2.01 (3)	0.652
Lungs diseases (%)	4.70 (7)	4.03 (6)	0.777
Hyperthyroidism (%)	2.68 (4)	2.01 (3)	0.702
Hypothyroidism (%)	3.36 (5)	4.70 (7)	0.556
Hypercholesterolemia (%)	5.37 (8)	8.05 (12)	0.354
**Current Medications**			
ASA (%)	26.17 (39)	12.08 (18)	0.002
β -blockers (%)	14.09 (21)	15.44 (23)	0.744
ACEI (%)	18.12 (27)	10.74 (16)	0.070
AT_2_-blockers (%)	2.68 (4)	0 (0)	0.044
Calcium channel blockers (%)	6.04 (9)	5.37 (8)	0.803
Diuretics (%)	9.4 (14)	8.05 (12)	0.681
Steroids (%)	2.01 (3)	0.67 (1)	0.314
Antidiabetic therapy (%)	4.03 (6)	2.68 (4)	0.520
Insulin (%)	1.34 (2)	2.68 (4)	0.409
Heparin (%)	0.67 (1)	0.67 (1)	0.999
Anticoagulants (%)	5.37 (8)	6.04 (9)	0.803
Nitrates (%)	0.67 (1)	0 (0)	0.316
Statins (%)	8.72 (13)	2.68 (4)	0.025
**Artery Sizes**			
Mean ICA diameter ± SD (mm)	4.01 ± 1.01	4.07 ± 1.18	0.617
C6 segment diameter ± SD (mm)	3.77 ± 0.88	3.99 ± 0.82	0.023
C7 segment diameter ± SD (mm)	2.88 ± 0.77	2.95 ± 0.67	0.392
**Tortuosity Descriptors**			
Relative Length ± SD	0.46 ± 0.19	0.51 ± 0.17	0.023
Sum of Angle Metrics ± SD	0.39 ± 0.21	0.32 ± 0.21	0.003
Product of Angle Distance ± SD	0.38 ± 0.19	0.32 ± 0.21	0.011
Triangular Index ± SD	0.30 ± 0.11	0.27 ± 0.14	0.034
Inflection Count Metric ± SD	0.30 ± 0.16	0.22 ± 0.12	<0.001

ICA, internal carotid artery; SD, standard deviation; ASA, acetylsalicylic acid; ACEI, angiotensin-converting-enzyme inhibitors; AT_2_-blockers, Angiotensin II receptor blockers.

**Table 2 jcm-08-00237-t002:** Association of intracerebral aneurysm development risk factors with internal carotid tortuosity descriptors.

Tortuosity Descriptor	Women (*n* = 227)	Men (*n* = 71)	*p*-Value
Relative Length ± SD	0.49 ± 0.19	0.48 ± 0.18	0.812
Sum of Angle Metrics ± SD	0.37 ± 0.21	0.31 ± 0.21	0.028
Product of Angle Distance ± SD	0.37 ± 0.20	0.30 ± 0.20	0.016
Triangular Index ± SD	0.30 ± 0.13	0.25 ± 0.10	0.006
Inflection Count Metric ± SD	0.27 ± 0.15	0.23 ± 0.13	0.046
	**Hypertension** **(*n* = 142)**	**No hypertension** **(*n* = 156)**	
Relative Length ± SD	0.49 ± 0.18	0.48 ± 0.19	0.900
Sum of Angle Metrics ± SD	0.37 ± 0.22	0.35 ± 0.20	0.327
Product of Angle Distance ± SD	0.36 ± 0.21	0.35 ± 0.19	0.543
Triangular Index ± SD	0.30 ± 0.14	0.28 ± 0.11	0.210
Inflection Count Metric ± SD	0.27 ± 0.16	0.25 ± 0.13	0.204
	**Diabetes Mellitus** **(*n* = 32)**	**No diabetes Mellitus** **(*n* = 266)**	
Relative Length ± SD	0.47 ± 0.16	0.49 ± 0.19	0.573
Sum of Angle Metrics ± SD	0.35 ± 0.21	0.36 ± 0.21	0.811
Product of Angle Distance ± SD	0.35 ± 0.21	0.35 ± 0.20	0.953
Triangular Index ± SD	0.30 ± 0.17	0.28 ± 0.12	0.372
Inflection Count Metric ± SD	0.25 ± 0.13	0.26 ± 0.15	0.690
	**Smoking** **(*n* = 38)**	**No Smoking** **(*n* = 260)**	
Relative Length ± SD	0.51 ± 0.18	0.48 ± 0.19	0.312
Sum of Angle Metrics ± SD	0.37 ± 0.21	0.36 ± 0.21	0.758
Product of Angle Distance ± SD	0.38 ± 0.21	0.35 ± 0.20	0.379
Triangular Index ± SD	0.27 ± 0.11	0.29 ± 0.13	0.324
Inflection Count Metric ± SD	0.26 ± 0.13	0.26 ± 0.15	0.989
	**Hypercholesterolemia** **(*n* = 20)**	**No Hypercholesterolemia** **(*n* = 278)**	
Relative Length ± SD	0.52 ± 0.17	0.48 ± 0.19	0.403
Sum of Angle Metrics ± SD	0.31 ± 0.20	0.36 ± 0.21	0.268
Product of Angle Distance ± SD	0.31 ± 0.20	0.36 ± 0.20	0.336
Triangular Index ± SD	0.26 ± 0.11	0.29 ± 0.13	0.295
Inflection Count Metric ± SD	0.23 ± 0.11	0.26 ± 0.15	0.360
	**History of Heart Attack** **(*n* = 5)**	**No History of Heart Attack** **(*n* = 293)**	
Relative Length ± SD	0.48 ± 0.14	0.48 ± 0.19	0.964
Sum of Angle Metrics ± SD	0.40 ± 0.32	0.36 ± 0.21	0.658
Product of Angle Distance ± SD	0.43 ± 0.37	0.35 ± 0.20	0.379
Triangular Index ± SD	0.26 ± 0.08	0.29 ± 0.13	0.653
Inflection Count Metric ± SD	0.21 ± 0.07	0.26 ± 0.14	0.447
	**History of Ischemic Stroke (*n* = 28)**	**No History of Ischemic Stroke (*n* = 270)**	
Relative Length ± SD	0.48 ± 0.16	0.48 ± 0.19	0.883
Sum of Angle Metrics ± SD	0.36 ± 0.22	0.36 ± 0.21	0.999
Product of Angle Distance ± SD	0.33 ± 0.23	0.36 ± 0.20	0.545
Triangular Index ± SD	0.30 ± 0.19	0.29 ± 0.12	0.582
Inflection Count Metric ± SD	0.25 ± 0.14	0.26 ± 0.14	0.724
	**History of Subarachnoid Hemorrhage (*n* = 18)**	**No history of Subarachnoid Hemorrhage (*n* = 280)**	
Relative Length ± SD	0.53 ± 0.18	0.48 ± 0.19	0.284
Sum of Angle Metrics ± SD	0.42 ± 0.22	0.35 ± 0.21	0.176
Product of Angle Distance ± SD	0.46 ± 0.22	0.35 ± 0.20	0.024
Triangular Index ± SD	0.30 ± 0.08	0.29 ± 0.13	0.733
Inflection Count Metric ± SD	0.25 ± 0.13	0.26 ± 0.15	0.828

**Table 3 jcm-08-00237-t003:** Correlation of tortuosity descriptors and continuous variables.

Variable	RL	SOAM	PAD	TI	ICM
Age (years)	0.053	0.016	0.002	0.089	0.024
*p*-Value	0.371	0.791	0.973	0.131	0.682
C6 segment diameter (mm)	−0.040	−0.070	−0.084	−0.082	−0.016
*p*-Value	0.498	0.232	0.155	0.162	0.787
C7 segment diameter (mm)	−0.026	0.024	0.025	0.045	0.068
*p*-Value	0.659	0.683	0.668	0.444	0.245
Mean ICA diameter (mm)	−0.187	−0.034	−0.057	0.072	0.091
*p*-Value	0.001	0.562	0.335	0.219	0.123
MCA diameter (mm)	0.082	−0.003	0.024	−0.031	−0.013
*p*-Value	0.162	0.965	0.680	0.596	0.827
ACA diameter (mm)	0.024	0.026	0.060	−0.028	0.034
*p*-Value	0.681	0.663	0.309	0.635	0.568
Time after SAH (months)	−0.437	0.447	0.229	0.534	0.773
*p*-Value	0.118	0.109	0.432	0.049	0.001

RL, Relative Length; SOAM, Sum of Angle Metrics; PAD, Product of Angle Distance; TI, Triangular Index; ICM, Inflection Count Metrics; ICA, internal carotid artery; MCA, middle cerebral artery; ACA, anterior cerebral artery; SAH, subarachnoid hemorrhage.

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
