# Peer review of "Tortuosity of the Internal Carotid Artery and Its Clinical Significance in the Development of Aneurysms"

_jcm, 2019, doi:10.3390/jcm8020237_

Reviewer 1 Report

please better report the patient selection and type of patient included in the present dataset.

did many of these patients have a dissection in their history ?

as this is a cohort study with a single meaurement we do not know the relaltion between tortuosity and development of carotid aneurysm. as such , the conclusion of the authors that tortuosity might reuslt in aneurysm development is premature. The authors should merely report "co-prevalence" instead of a causal relationship which can never be determined with the use this methodology. Recently there was a publication on the co-prevalence of extracranial carotid aneurysms in patients with proven intracrnial aneurysms. this co-prevalence may have an interrelationship with tortuosity. See Pourier VEC et al. PLoS One 2017 PMID 29131823

please better differentiate between EXTRACRANIAL tortuosity versus INTRACRANIAL tortuosity; and likewise between EXTRACRANIAL carotid aneurysm versus intracranial aneurysm.

although the methods of arterial tracking and definiions of parameters have been published elsewhere in detail, it would truly help the reader if the authors could add a figure / drawing with all the measures visualized shortly.

please better define your primary endpoint ersus secondary endpoints of this study and report these within the Methods sections

report the Results in the same order for primary vs secondary endpoints.

patients with SAH in relation to tortuosity corrected for age effect ?

how many observers performed the measurements ? please report intra and inter observer agreement.

in this perspective, the authors might be interested to read the recent publication on software packages to compare tortuosity:  de Vries EE et al. Comparability of semiautomatic tortuosity measurements in the carotid artery. Neuroradiology 2018 PMID 30338348.

Author Response

1.     Please better report the patient selection and type of patient included in the present dataset.

@Additional informations about patients selection were added to Experimental Section

2.     Did many of these patients have a dissection in their history ?

@In our study there were no patients with ICA dissection in their history.

3.     as this is a cohort study with a single meaurement we do not know the relaltion between tortuosity and development of carotid aneurysm. as such , the conclusion of the authors that tortuosity might reuslt in aneurysm development is premature. The authors should merely report "co-prevalence" instead of a causal relationship which can never be determined with the use this methodology. 

@Conclusions were corrected to more applicable.

4.     please better differentiate between EXTRACRANIAL tortuosity versus INTRACRANIAL tortuosity; and likewise between EXTRACRANIAL carotid aneurysm versus intracranial aneurysm.

@All of the measurements concerned only intracranial segments of ICA. Similarly, in our study we analysed only patients with intracranial ICA aneurysms. Adequate sentence was added to Experimental Section.

5.     although the methods of arterial tracking and definiions of parameters have been published elsewhere in detail, it would truly help the reader if the authors could add a figure / drawing with all the measures visualized shortly.

@Figure visualizing parameters was added to manuscript.

6.     please better define your primary endpoint versus secondary endpoints of this study and report these within the Methods sections

@Primary and secondary endpoints were defined in Experimental section.

 7.     report the Results in the same order for primary vs secondary endpoints.

@Results were reported in the same order for primary and secondary endpoints.

8.     patients with SAH in relation to tortuosity corrected for age effect ?

@As In our study there was no significant correlation between ICA tortuosity and age we did not performed correction for age effect.

9.     how many observers performed the measurements ? please report intra and inter observer agreement.

@As measurements in our study are performed automatically by the software, presence of an observers and an observer agreements are not applicable.

Reviewer 2 Report

The English needs addressing in terms of the "tenses" of verbs like "is," "was," and were.  I also suggest including a figure that shows an example angiogram with all of the derived measurements. That way the reader knows what the authors are talking about without having to look up their previous manuscript.

I also think that mention of how the finding might relate to aneurysm formation would interesting, and how fluid mechanics with vessel trotuosity might influence fluid fow.  Why is vessel tortuosity related to aneurysm formation?

Author Response

1.     The English needs addressing in terms of the "tenses" of verbs like "is," "was," and were.

@English was corrected by a native speaker.

2.     I also suggest including a figure that shows an example angiogram with all of the derived measurements.

@Appropriate figure was added to manuscript.

3.     I also think that mention of how the finding might relate to aneurysm formation would interesting, and how fluid mechanics with vessel trotuosity might influence fluid fow.  Why is vessel tortuosity related to aneurysm formation?

@A possible explanations of tortuosity influence on aneurysms formation were given in second paragraph of discussion, together with characteristics of blood flow in tortuous arteries (from Rikhtegar et al. study). Any further explanations require more extensive research of hemodynamics in tortuous arteries, which haven’t been dome so far.

Reviewer 3 Report

This is an interesting retrospective study examining whether ICA tortuosity is associated with ICA aneurysm formation. The main limitation of the study has been partly addressed by the authors in the discussion section of the manuscript (whether it is the tortuosity that causes the aneurysm or the other way round). An aneurysmal artery expands in all three dimensions, including its length, so some tortuosity in inevitable.

Apart from that, I would like to see a comment about:

the finding that ICA aneurysms are less frequent in patients with ischemic strokes

the finding that alcoholism protects from ICA aneurysms

Author Response

1.     Apart from that, I would like to see a comment about: the finding that ICA aneurysms are less frequent in patients with ischemic strokes

@Patients with history of ischemic stroke are more likely to take antihypertensive drugs and statins, which might prevent from aneurysm formation. As that finding is not strictly related to our study main subject we decided not to address it in the discussion.

2.     the finding that alcoholism protects from ICA aneurysms

@Due to small number of alcoholics in our study group we believe that such finding is more a statistical phenomenon, therefore we decided not to address it in the discussion.